# The Effects of Resveratrol in the Treatment of Metabolic Syndrome

**DOI:** 10.3390/ijms20030535

**Published:** 2019-01-28

**Authors:** Chih-Yao Hou, You-Lin Tain, Hong-Ren Yu, Li-Tung Huang

**Affiliations:** 1Department of Seafood Science, National Kaohsiung University of Science and Technology, Kaohsiung 811, Taiwan; chihyaohou@gmail.com; 2Institute for Translational Research in Biomedicine, Kaohsiung Chang Gung Memorial Hospital and Chang Gung University College of Medicine, Kaohsiung 833, Taiwan; tainyl@hotmail.com (Y.-L.T.); yuu2004taiwan@yahoo.com.tw (H.-R.Y.); 3Department of Pediatrics, Kaohsiung Chang Gung Memorial Hospital and Chang Gung University College of Medicine, Kaohsiung 833, Taiwan; 4Department of Traditional Medicine, Chang Gung University, Linkow 333, Taiwan

**Keywords:** resveratrol, metabolic syndrome, high-fat diet, resveratrol derivatives

## Abstract

Resveratrol, also known as 3,5,4′-trihydroxystilbene, is a natural polyphenol that occurs as a phytoalexin. It is produced by plant sources such as grapes, apples, blueberries, plums, peanuts, and other oilseeds. This compound has a variety of effects on human health and diseases. This review summarizes the mounting evidence that resveratrol is helpful in treating metabolic syndrome and related disorders. Resveratrol can be provided either early as a reprogramming agent or later as part of treatment. A few of the main mechanisms underlying the beneficial effects of resveratrol on metabolic syndrome are outlined. This review also discusses the potential of resveratrol derivatives as a complementary or alternative medicine. In conclusion, resveratrol could be a useful regimen for the prevention and treatment of metabolic syndrome and its related conditions.

## 1. Introduction

Resveratrol (*trans*-3,5,4′-trihydroxystilbene) is widely known as a phenolic compound from the stilbene family, with a C6–C2–C6 structure containing three hydroxyl groups and powerful antioxidant activity, which is found in various plants. In plants, resveratrol serves as an ultraviolet (UV) protectant and plays a role as a defense compound against pathogens infection, injury, and abiotic stresses [1,2]. It is present in over 70 types of plants, including grape skins, grape seeds, giant knotweed, cassia seeds, passion fruit, white tea, plums, and peanuts [2,3,4,5]. 

Resveratrol was first isolated from the roots of the white hellebore (*Veratrum grandiflorum* O. Loes). In 1963, it was found in the roots of *Polygonum cuspidatum*. Initially characterized as a phytoalexin, resveratrol attracted little interest until 1992 [6]. It has also been found in Japanese Knotweed [7], grapevine tissue and barriers and in grapevine cell cultures under abiotic and biotic stresses [4,6]. Due to resveratrol’s utility, a sustainable source of resveratrol is required. Currently, the majority of commercial resveratrol is extracted from *Polygonum cuspidatum* [3], leading to a variety of compositions and purities. The requirement for lower cost and high purity resveratrol has promoted its heterologous biosynthesis in engineered microbes. The first study on the quantitative production of resveratrol was in *Saccharomyces cerevisiae* based on feeding ρ-coumaric acid and the expression of two heterologous enzymes 4-coumaryl-CoA ligase (4CL) and resveratrol synthase (VST1, also called stilbene synthase, STS) [3,8]. 

Resveratrol has been used in medicines, dietary supplements and as a functional food ingredient; positive effects of resveratrol on health have also been confirmed in several studies. It has preventive effects on oxidation, inflammation, cardiovascular disease and has anticarcinogenic activity, among other health benefits [5,9,10,11,12]. 

## 2. The Synthesis and Functions of Resveratrol

A review of literature has shown that resveratrol is synthesized by plants as a response to unfavorable or stressful conditions, damage, and mechanical injury. In the physiological metabolism of plants, glucose is metabolized to 4-coumaroyl-CoA and is combined with malonyl-CoA via stilbene synthase to produce trans-resveratrol (Figure 1). In response to bacteria or fungi or UV exposure, two forms of resveratrol can be produced, namely trans- and cis-isomers. In addition, while trans-resveratrol occurs naturally in grapes, various technologies have identified cis-resveratrol and its glucoside in wines of diverse origin. It was detected that the vinification process transforms some *trans*-resveratrol into its *cis*-form. Nevertheless, when protected from light, the *trans*-form can be stable for months, except in high pH, while cis-resveratrol was stable only in neutral pH [13]. In addition, resveratrol has been demonstrated to be a natural antimicrobial agent [14]. It also modulates gut microbiota composition and hence has anti-diabetic, anti-obesity, and anti-atherosclerosis properties [15].

Resveratrol is most known for its purported health benefits (e.g., heart health-related claims about the resveratrol present in red wine). Sources of resveratrol include Japanese knotweed, red grape varieties and other berries, but it can also be produced synthetically and by fermentation. A proposed monograph on trans-resveratrol produced by yeast fermentation represents the first proposed authoritative standard for resveratrol as a food ingredient. No obvious resveratrol toxicity has been noted in humans [16]; however, *trans*-resveratrol at a dose of 2000 mg twice daily caused diarrhea and clinically irrelevant changes of serum potassium and total bilirubin levels in healthy subjects [17].

## 3. Metabolic Syndrome

Metabolic syndrome is a cluster of metabolic disorders. The condition includes hypertension, high blood sugar or diabetes mellitus, obesity, excess body fat around the waist, and abnormal cholesterol or triglyceride levels, patients presenting with several of these conditions have a greater chance for future cardiovascular events than those with any one factor alone. Metabolic syndrome is a growing problem globally and is a contributor to non-communicable diseases such as type II diabetes, cardiovascular disease, and cognitive deficits. A large body of epidemiologic literature supports an inverse relationship between birth weight and both systolic blood pressure and hypertension. In addition, hypertension can originate from early-life insults, referred to as the “developmental origins of health and disease (DOHaD)’’, by Gluckman et al., who proposed the concept of DOHaD after observing the enduring effects of the fetal environment on physical health and disease in adulthood [18]. The process of fetal programming or developmental plasticity is one of the core assumptions of DOHaD [19]. Metabolic syndrome is now considered a disease of developmental origin [18], which has become a main priority of recent research.

## 4. Mechanisms of Resveratrol Action in the Treatment of Metabolic Syndrome

Resveratrol acts through different mechanisms to improve the symptoms of metabolic syndrome and related disorders [20]. Below, we summarize a few of the main targets of resveratrol in healthy and disease states.

### 4.1. SIRT1

The sirtuin family of proteins catalyzes the Nicotinamide adenine dinucleotide (NAD^+^)-dependent deacylation of acyl-lysine residues. Changes in sirtuin expression are critical in several diseases, including metabolic syndrome, cardiovascular diseases, cancer, and neurodegeneration. Increasing levels of the SIRT1 protein help control some disease related conditions including obesity, cardiovascular diseases and neurodegeneration [21]. Resveratrol activates Sirtuins (SIRT)1, which has a variety of biological and pharmacological effects, including cardio protective, antioxidant, and anti-inflammatory effects [22,23]. SIRT1 transgenic mice display some features resembling calorie restriction.

### 4.2. AMPK

AMP-activated protein kinase (AMPK) is a protein kinase that maintains homeostasis in cells and organs. Interestingly, the health benefits of SIRT1 activation following resveratrol overlap in many ways with those conferred by AMPK activation. AMPK has emerged as a key nutrient sensor, with the ability to regulate whole-body metabolism [24], coordinating many metabolic reactions [25] to maintain metabolic homeostasis and induce the inflammatory response [26]. Tamaki et al. reported that resveratrol has the ability to activate SIRT1, AMPK, and Nuclear factor (erythroid-derived 2)-like 2 (Nrf2)/antioxidant defense pathways in a rat periodontitis model [27]. 

### 4.3. Renin-Angiotensin System

RAS plays an important role in metabolic syndrome [28]. In Ang II-stimulated vascular smooth muscle cells (VSMCs), resveratrol treatment markedly decreased the number of senescence associated β-galactosidase stained cells and pro-fibrotic protein expression and increased the expression of AT2R and MasR [29]. 

### 4.4. Others

Recently, an important feature of resveratrol has been documented. Resveratrol may affect gut microbiota and their metabolic products, such as short chain fatty acids and intraluminal lipids, and hence alleviate metabolic syndrome [15,30]. Resveratrol also modulates mitochondrial function and dynamics and may induce the regulation of transcription factors that activate or repress the expression of mitochondria-related genes, causing alterations in the mitochondrial physiology [31]. A wide range of SIRT1 substrates have already been described, including key regulators of mitochondrial respiration, lipid metabolism and aging processes, such as the Forkhead box class O (FOXO) family of transcription factors [32]. In addition to its activation of SIRT1, resveratrol is presumably involved in transcriptional regulation of the nuclear factor kappa-B cells (NF-κB) family of proteins. Nrf2-Keap1 signaling also has been reported in diabetic nephropathy in response to resveratrol [33]. Anti-inflammation is also one of the major mechanisms of action by which resveratrol acts on metabolic syndrome [34].

## 5. Treatment of Metabolic Syndrome

The treatment of metabolic syndrome consists of mainly life style changes and pharmacological approaches. Since metabolic syndrome is now considered to have a developmental origin, the first approach should be reprogramming [18]. Later in life, resveratrol intervention can be considered as a therapeutic supplement [35]. Figure 2 outlines the possible underlying mechanisms of resveratrol effects on metabolic syndrome and therapeutic strategies including early-life reprogramming and later-life intervention.

### 5.1. Reprogramming of Metabolic Syndrome by Resveratrol

Metabolic syndrome is defined as central obesity plus any two of the following factors: elevated triglycerides, reduced High-density lipoproteins (HDL) cholesterol, elevated blood pressure or blood glucose abnormalities [36]. Metabolic syndrome is now considered a disease of developmental origin [18]. Developmental programming, also known as DOHaD [37], refers to exposure during prenatal or early postnatal development that may cause permanent changes in an individual’s physiology and metabolism that predispose them to non-communicable diseases, such as Alzheimer’s, asthma, chronic kidney disease, and diabetes mellitus [38]. Reprogramming offers an opportunity, in the critical period of early development, to reverse this predisposition [39,40]. Tain et al. have previously reported on the potential role of melatonin in reprogramming metabolic syndrome following early perinatal insults. Reprogramming metabolic syndrome following early perinatal insults by resveratrol has also been reported. Signaling during reprogramming include SIRT1, Nrf2, and AMPK, all of which can be adjusted by resveratrol [23,27,41]. We therefore in this review article discuss the role of resveratrol in the treatment of metabolic syndrome. 

The effective treatment of metabolic syndrome has been less convincing in human studies than in animal studies [42]. We will first discuss human studies, followed by preclinical studies on the effects of resveratrol in metabolic syndrome.

### 5.2. Treatment of Metabolic Syndrome by Resveratrol

#### 5.2.1. Clinical Studies

Resveratrol is being tested in many clinical trials, as it may exert these effects by targeting several main metabolic sensor/effector proteins, such as AMPK, SIRT1, and Peroxisome proliferator-activated receptor gamma coactivator-1α (PGC-1α) [43]. A randomized, double-blind, placebo-controlled clinical trial in 24 patients with diagnoses of metabolic syndrome showed that resveratrol significantly decreases weight, Body Mass Index (BMI), fat mass, waist circumference, area under curve of insulin, and total insulin secretion [44]. In a randomized, four-month study of middle-aged men with metabolic syndrome, Korsholm and colleagues found that resveratrol treatment reduced sulfated androgen precursors in blood, adipose tissue, and muscle tissue, and increased these metabolites in urine [45]. They also found increased intracellular glycerol and an accumulation of long-chain saturated, monounsaturated, and polyunsaturated free fatty acids in the muscle of resveratrol-treated men. In addition, changes in urinary derivatives of aromatic amino acids, suggested an altered gut microbiota composition following resveratrol treatment. They concluded that there was evidence that resveratrol has subtle but robust effects on several metabolic pathways in men with metabolic syndrome. They stress that the effects would not have been detected by routine analyses. Amiot et al. reviewed the clinical evidence of chronic food or nutrient supplements in patients with metabolic syndrome, concluding that diets rich in polyphenols, for example the Mediterranean diet, could be an effective nutritional strategy [46]. Recently, a meta-analysis investigating inflammatory markers among patients with metabolic syndrome and related disorders showed that resveratrol supplementation significantly decreased C-reactive protein and tumor necrosis factor-α concentrations, while there was no significant change in Interleukin 6 and superoxide dismutase concentrations [47]. They concluded that resveratrol supplementation showed a promising effect on lowering some inflammatory markers among patients with metabolic syndrome and related disorders [48]. Longevinex, a modified resveratrol, improved endothelial function in patients with metabolic syndrome [49]. Longevinex specifically improved endothelial function in 34 patients with metabolic syndrome who were receiving standard therapy for lifestyle-related diseases [49]. In contrast, 1000 or 150 mg daily resveratrol in middle-aged, community-dwelling men with metabolic syndrome showed negative results [50]. Resveratrol treatment did not improve their inflammatory status, glucose homeostasis, blood pressure, or hepatic lipid content. However, resveratrol significantly increased total cholesterol, LDL cholesterol, and fructosamine levels compared with controls [50]. Collectively, the results of clinical studies are unsatisfactory and varied due to different study designs, resveratrol dosages, and treatment durations, or low bioavailability of the supplement [51]. 

#### 5.2.2. Preclinical Studies

● Brain

Resveratrol might alter gut microbiome, affect the gut-brain axis, and have a role in brain disorders [15,30,52]. Aquaporins are integral, transmembrane, water channel proteins that are now implicated in metabolic syndrome [51]. Resveratrol also exerts beneficial effects by modulating brain aquaporins and exert beneficial effects [53]. Evidence suggests that resveratrol may augment brain-derived neurotrophic factor (BDNF) synthesis, which can prevent bisphenol-A-induced autism, metabolic syndrome, and diabetes mellitus [23]. People with metabolic syndrome are more likely to experience cognitive dysfunction as obesity is an independent risk factor for the development of cognitive decline in both the middle-aged and elderly [48,54]. Resveratrol might also be beneficial for the prevention of diabetes-related cognitive impairment [55]. In streptozotocin-injected diabetic rats, resveratrol administration for four weeks improved hippocampus-dependent cognitive deficit. Heyward et al. showed that mice maintained on a high-fat diet develop impaired hippocampus-dependent spatial memory possibly mediated by the epigenetic dysregulation of SIRT1 within the hippocampus [56]. Resveratrol also reversed changes to caspase-3, Bax, Bcl-2, *N*-Methyl-d-Aspartate (NMDA) receptor, and brain derived neurotrophic factor activity in the hippocampus [57].

Our research group found that a combined maternal and postnatal high-fat diet caused metabolic syndrome and cognitive dysfunction in adult rat offspring [58]. We observed lower expression of dorsal hippocampal BDNF in maternal and postnatal high-fat diet rats, which was reversed by resveratrol therapy, as was the associated cognitive dysfunction [58]. Rats with diet-induced metabolic syndrome also had less dorsal hippocampal SIRT1, which resveratrol was also able to restore [59]. Our research also found lower dorsal hippocampal AT1R expressions in diet induced metabolic syndrome rats, which again was restored by resveratrol treatment [59]. These findings suggest that resveratrol could reverse cognitive dysfunction through modulating dorsal hippocampal BDNF, SIRT1, and AT1R in rats with metabolic syndrome. 

● Liver

Lipid metabolism disorders are an important predisposing factor for fatty liver disease pathogenesis, which is characterized by excessive lipid accumulation in the liver [60,61]. SIRT1 is the most extensively studied protein implicated in lipid metabolism disorders, and plays a role in both alcoholic and nonalcoholic fatty liver disease (NAFLD) [41]. SIRT1 plays multiple roles including regulating hepatic lipid metabolism, controlling hepatic oxidative stress, and inhibiting hepatic inflammation through deacetylating transcriptional regulators that promote the progression of fatty liver disease. SIRT1 could protect against a high-fat diet or alcohol-induced hepatic steatosis via various metabolic pathways [41,62,63]. In rats, *N*′-Nitrosodimethylamine-induced liver fibrosis was alleviated by resveratrol. Resveratrol was able to revert the liver damages by increasing histological integrity, repressing oxidative damage and inhibiting liver stellate cells activation by down regulating α-Smooth muscle actin expression [64]. A study of the effects of resveratrol treatment on hepatic oxidative stress in a rat model of metabolic syndrome induced by high fructose diet showed that resveratrol therapy had beneficial effects on liver paraoxonase activity [65]. Similarly, resveratrol showed beneficial effects on hepatic oxidative stress in a rat model of fructose diet-induced metabolic syndrome [66]. Burgess et al. tested resveratrol in pigs with high calorie and high-fat/cholesterol-induced metabolic syndrome [67]. They found that supplemental resveratrol positively influenced glucose metabolism pathways in the liver and skeletal muscle and led to improved glucose control.

Studies from our group demonstrated mRNA levels of angiotensinogen, renin, ACE1, and AT1R in the liver were increased in rats exposed to a combined maternal and postnatal high-fat diet, which were significantly decreased following resveratrol therapy. In addition, histopathology showed that resveratrol therapy also decreased lipid droplets, MDA levels, and leptin receptors in the liver of rats with diet induced metabolic syndrome [68]. These findings suggest that resveratrol therapy might play a beneficial role in the liver in rats with metabolic syndrome via renin-angiotensin system modulation [68].

● Kidney

Diabetic nephropathy is a common complication of metabolic syndrome. Kim et al. showed that resveratrol prevents diabetic nephropathy in db/db mice through the phosphorylation of AMPK and activation of SIRT1-PGC-1α signaling, which appears to prevent lipotoxicity-related apoptosis and oxidative stress in the kidney [69].

Emerging evidence suggests an active role for the renin-angiotensin system (RAS) in the development of hypertension. Our research group found that a combination of a maternal and a postnatal high-fat diet increased the level of Ang I and reduced the level of Ang (1–7) in plasma, resveratrol treatment therapy significant reduced plasma Ang II levels and increased AMPK2α phosphorylation [62] and prevented Ang II induced-hypertension via AMPK activation [62,70]. In addition, the combination of a maternal and a postnatal high-fat diet decreased mRNA expression of Ulk1 and Atg5. Resveratrol treatment prevented the decreases in Ulk1 and Atg5 mRNA levels as well as increasing the LC3-II/LC3-I ratio [62]. These results indicate that a combined maternal and post-weaning high-fat diet can inhibit autophagy, which was rescued by resveratrol. The above findings suggest that resveratrol could reverse nephropathy and hypertension in rats with metabolic syndrome through the regulation of RAS, oxidative stress, autophagy, and nutrient-sensing signals.

● Adipose tissue

Excessive accumulation and persistence of low-grade inflammation in adipose tissue are typical features of obesity. Adipose tissue is recognized as a metabolically active organ and is involved in producing many adipokines to modulate target organs [71]. For example, leptin plays an important role in metabolic regulation, energy expenditure, and glucose homeostasis. The presence of high concentrations of leptin accompanied with down-regulation of the hypothalamic leptin receptors and dysfunction in leptin signaling together comprise “leptin resistance” in obesity [72]. Resveratrol treatment can decrease blood leptin concentrations [73] and seems to ameliorate leptin resistance. Adipose tissues store triglycerides and release fatty acids through lipogenesis and lipolysis, respectively. Resveratrol treatment has been demonstrated to decrease lipogenesis and increase lipolysis, thus having an anti-obesity effect [74]. It has also been shown to increases the capacity of thermogenesis in brown adipose tissue [75]. Resveratrol is stronger inhibitor of PAI-1 expression than PI3K, Sirt1, AMPK, ROS, and Nrf2, through NF-κB modulation in inflamed adipose tissue [76]. An in vitro study also showed that resveratrol treatment inhibited preadipocyte proliferation, adipogenic differentiation and inflammatory cytokines production in a Sirt1-dependent manner [77]. Other beneficial effects of resveratrol treatment on adipocytes include improving insulin signaling and mitochondrial dynamics [78,79]. Resveratrol plus quercetin upregulated white adipose tissue SIRT1 and SIRT2 in a rat model of sucrose diet-induced metabolic syndrome [80]. Recently it has been thought that the antiobesity effect of resveratrol is due to alteration in the gut microbiome and its related consequences [81]. It is also proposed that altered microbiome might promote brown adipose tissue activation and white adipose tissue browning [82].

We and colleagues also found that resveratrol can improve several dysregulated RAS molecules of visceral adipose tissue that resulted from prenatal and postnasal high-fat diet [59]. Since RAS of adipose tissue has profound systemic effects [83], resveratrol treatment has modulatory effects on the cardiovascular system through adipose tissue. The above findings suggest that resveratrol could affect adipose tissue and has beneficial effects in rats with metabolic syndrome. 

Obesity results in chronic inflammation in adipose tissue, and that cells of the innate immune system, particularly macrophages, are crucially involved in adipose inflammation [84]. Macrophages are subdivided into M1 and M2 in general. M1 macrophages are characterized by the production of pro-inflammatory cytokines, while M2 is anti-inflammatory macrophages that involved in the resolution of the inflammatory process and tissue repairing. The previous study has shown that resveratrol exerts anti-inflammatory effects by suppressing M1 microglia activation and promotes microglia polarization toward M2 phenotype via PGC-1α in conditions of neuroinflammatory injury [85]. While the role of resveratrol for macrophage polarization seems controversial for adipose tissue. Resveratrol has been shown to attenuate hypoxia-induced macrophage migration to visceral white adipose tissue [86]. However, resveratrol supplementation fails to improve obesity-induced macrophage infiltration and switching [87]. Thus, the effect of resveratrol on macrophage polarization in adipose tissue may be mechanism-dependent. 

● Pancreas and insulin sensitivity

Insulin resistance is a main feature of metabolic syndrome, and one for which resveratrol is especially effective. When Chen et al. treated insulin-resistant KKA(y) mice with 2 and 4 g/kg diets of resveratrol for 12 weeks [88], they found that resveratrol intervention reduced blood glucose and serum insulin levels, improved insulin and glucose tolerance, and increased serum adiponectin. They concluded that resveratrol could improve insulin sensitivity and ameliorate insulin resistance in KKA(y) mice [88]. Lagouge et al. investigated mice on a high-fat diet with or without resveratrol supplementation for 15 weeks. They found a significant reduction in fasting insulin levels in the resveratrol-treated rats compared with rats without resveratrol supplementation. This decrease in insulin was not accompanied by alterations in fasting glucose levels and was accompanied by improved insulin sensitivity [89].

Our research group previously reported that resveratrol could reverse the increased sugar area under curve and insulin area under curve in rats exposed to a combined maternal and postnatal high-fat diet, through increased levels of SIRT1 mRNA and expression in the pancreas. In addition, resveratrol could reduce ACE in rats exposed to a combined maternal and postnatal high-fat diet [59]. Moreover, the higher levels of plasma angiotensin II, indicating insulin resistance, seen in rats exposed to a combined maternal and postnatal high-fat diet was reversed by resveratrol treatment. The above findings suggest that resveratrol could reduce insulin resistance in rats with metabolic syndrome, in part through RAS regulation.

Table 1 summarizes main clinical and pre-clinical studies of resveratrol on metabolic syndrome.

## 6. Resveratrol Derivatives

Despite the therapeutic effects of resveratrol, its experimental application has limited application due to its poor solubility and low bioavailability. Recent studies have been aimed at designing novel resveratrol formulations to overcome its poor solubility. Encapsulation of resveratrol into nanodevices has been explored. Resveratrol cross-linked chitosan nanoparticles modified with phospholipids (RVC-lipid) were synthesized using a double emulsion technique, to produce an amorphous structure, with a mean particle size of 950 nm in ethanol, and an encapsulation efficiency of 63.82% in an aqueous medium and 85.59% in an ethanol medium. However, based on assays of antioxidant activity, the scavenging activity of RVC-lipid nanoparticles was inferior to that of resveratrol due to its prolonged release [91]. Elongation of the conjugated chain of resveratrol has proved to be an effective strategy to further improve its antioxidative capacity and radical scavenging ability. In addition, stable s-cis conformers of long chain resveratrol analogs are more favorable for the production of phenoxyl radicals and cation radicals than trans conformers, with concomitant enhancement in antioxidative activity [92]. Oh et al. reported on twelve resveratrol derivatives prepared using acyl chlorides of different chain lengths (C3:0–C22:6) and found that the resveratrol showed the highest antioxidant activity in oil-in water emulsion, while its derivatives (RC6:0, RC8:0, RC10:0, RC12:0, RC16:0) showed better antioxidant activity in a bulk oil system. On the other hand, resveratrol esters RC20:5n-3 (REPA) and RC22:6n-3 (RDHA) had the highest antioxidant activity when added to ground meat [9]. These results indicate that esterification had more antioxidant activity in the oil system and also demonstrated that the effect of esterification position, number of esterification substitution and the effect on bioactivity of esterification is still unclear and worth future investigation.

## 7. Conclusions

An enormous amount of clinical and animal studies on resveratrol suggests that it may improve health, prevent, and/or treat chronic diseases, such as metabolic syndrome. Given that life style modifications are difficult for the general population to adopt, resveratrol is worth exploring through more clinical and experimental studies. However, the effective treatment of metabolic syndrome has been less convincing in humans than in animal studies. Further studies are needed to understand the genetic factors that account for the differences in bioavailability and physiological responses to resveratrol among individuals. Moreover, it is important to develop different resveratrol derivatives in future studies. 

## Figures and Tables

**Figure 1 ijms-20-00535-f001:**
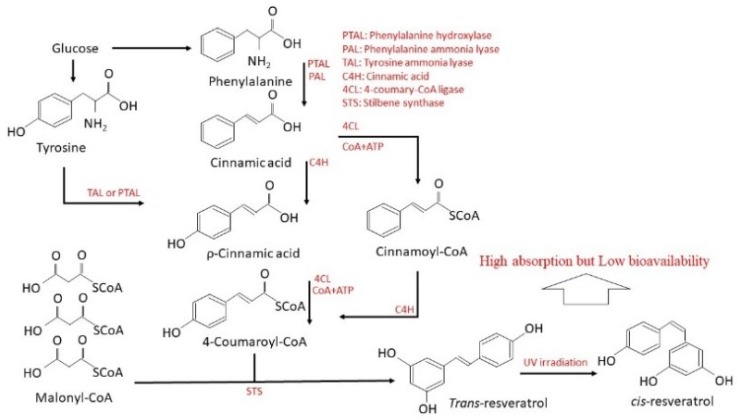
Pathway of resveratrol in plant.

**Figure 2 ijms-20-00535-f002:**
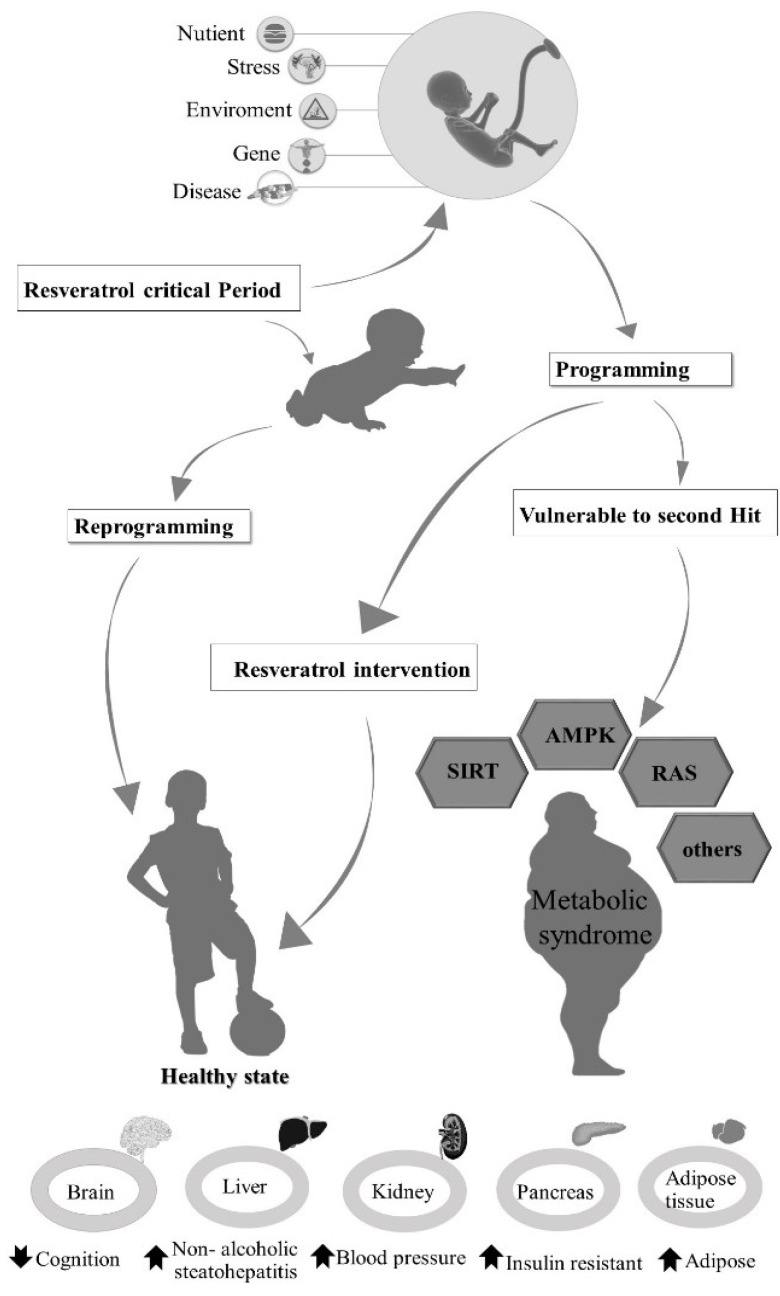
The role of resveratrol in metabolic syndrome.

**Table 1 ijms-20-00535-t001:** Summary of clinical and animal studies of resveratrol in metabolic syndrome.

Aim	Treatment	Result	References
Human			
Modified resveratrol Longevinex improves endothelial function in adults with metabolic syndrome receiving standard treatment	Thirty-four patients who had been treated for MetS and lifestyle-related disease were randomly assigned to group A and B. A: Longevinex was administered for 3 months and then discontinued for 3 months. B: Longevinex was administered between 3 and 6 months. (The patients received 1 capsule of Longevinex containing 100 mg trans resveratrol daily after dinner).	Flow-mediated dilatation significantly increased during the administration of Longevinex but decreased to baseline 3 months after the discontinuation of Longevinex in the group A patients. Conversely, in the group B patients, flow-mediated dilatation remained unchanged for the first 3 months without Longevinex but was significantly increased 3 months after the treatment with Longevinex.	[49]
Effects of Resveratrol on the Metabolic Syndrome	Middle-aged community-dwelling men (*n* = 74) with MetS, 66 of whom completed all visits (age, 49.5 ± 0.796 years). Daily oral supplementation with 1000 mg RSV (RSVhigh), 150 mg RSV, or placebo, for 16 weeks.	RSVhigh treatment significantly increased total cholesterol (*p* < 0.002), low-density lipoprotein (LDL) cholesterol (*p* < 0.006), and fructosamine (*p* < 0.013) levels compared with placebo.	[50]
Animal			
The effects of resveratrol on hepatic oxidative stress in metabolic syndrome model induced by high fructose diet	Male adult rats were induced by MetS, fructose solution (20% in drinking water). Resveratrol (10 mg/kg/day) was given by oral gavage.	Fructose-fed rats displayed statistically significant increases in TOS levels and decrease in PON activity compared to the control group, and prevented the decrease in liver PON activity caused by fructose.	[65]
Resveratrol improves oxidative stress and prevents the progression of periodontitis via the activation/antioxidant defense pathways in a rat periodontitis model	Animals in the periodontitis group underwent ligature-induced periodontitis and were given pure water. The melinjo resveratrol was administered orally at a dose of 10 mg/kg body weight.	Oral administration of melinjo resveratrol may prevent the progression of ligature-induced periodontitis and improve systemic oxidative and nitrosative stress.	[27]
The Combination of Resveratrol and Quercetin Attenuates Metabolic Syndrome in Rats	Each group of rats (control or MetS) received orally in drinking water or sucrose solution a mixture of RSV and QRC daily for 4 week. (1) RSV (10 mg/kg/day) + QRC (0.19 mg/Kg/day). (2) RSV (50 mg/kg/day) + QRC (0.95 mg/kg/day).	RSV + QRC administration improved the serum health parameters modified by MetS and upregulate SIRT 1 and SIRT 2 expression in white abdominal tissue in MetS animals.	[80]
Attenuation of insulin resistance, metabolic syndrome and hepatic oxidative stress by resveratrol in fructose-fed rats	Fructose-fed insulin resistant group (Dia) animals were fed 65% fructose for a period of 8 weeks, whereas control group animals were fed 65% cornstarch. Resveratrol, 10 mg/kg/day (Dia+Resv) or metformin 300 mg/kg/day (Dia+Met) were administered orally to the 65% fructose-fed rats.	Attenuation of hepatic oxidative stress in fructose-fed rat liver after resveratrol administration was associated with significant increase in the nuclear level of NRF2, and more effective than metformin in improving insulin sensitivity and attenuating metabolic syndrome and hepatic oxidative stress in fructose-fed rats.	[66]
A high-fat diet, leads to metabolic syndrome with spatial learning and memory deficits: beneficial effects of resveratrol	Receive high-fat hypercaloric diet. A therapeutic group with resveratrol on maternal high-fat diet/postnatal high-fat diet was raised for comparison (HHR). 50 mg/kg/day, for 16 weeks.	Treatment with resveratrol is able to rescue that maternal obesity/high-fat diet interacts with a postnatal high-fat diet to induce features of metabolic syndrome include alter biochemical profiles in the dorsal hippocampus, and lead to cognitive deficits.	[58]
Combined maternal and postnatal high-fat diet leads to metabolic syndrome and is effectively reversed by resveratrol: a multiple-organ study	Receive high-fat hypercaloric diet. Resveratrol 50 mg/kg/day, for 16 weeks.	Resveratrol ameliorated most of the features of metabolic syndrome and molecular alterations.	[59]
Resveratrol ameliorates maternal and post-weaning high-fat diet-induced nonalcoholic fatty liver disease via renin-angiotensin system	Receive high-fat hypercaloric diet. Resveratrol 50 mg/kg/day for offspring from post-weaning to postnatal day (PND) 120.	Resveratrol administration mediated a protective effect on rats on HF/HF by regulating lipid metabolism, reducing oxidative stress and apoptosis, restoring nutrient-sensing pathways by increasing Sirt1 and leptin expression, and mediating the renin-angiotensin system (RAS) to decrease angiotensinogen, renin, ACE1, and AT1R levels and increased ACE2, AT2R and MAS1 levels compared to those in the OHF group.	[68]
Improving glucose metabolism with resveratrol in a swine model of metabolic syndrome through alteration of signaling pathways in the liver and skeletal muscle	The swine developed metabolic syndrome by consuming a high-calorie, high–fat/cholesterol diet for 11 weeks. A hypercholesterolemic group diet with supplemental resveratrol (100 mg/kg/day).	Immunoblotting in the liver showed increased levels of mammalian target of rapamycin, insulin receptor substrate 1, and phosphorylated AKT in the HCRV group. Immunoblotting in skeletal muscle tissue demonstrated increased glucose transporter type 4 (Glut 4), peroxisome proliferating activation receptor coactivator 1α, peroxisome proliferator-activated receptor α, peroxisome proliferator-activated receptor, and phosphorylated AKT at threonine 308 expression as well as decreased retinol binding protein 4 in the HCRV group.	[67]
Investigating the Effects of Resveratrol on Chronically Ischemic Myocardium in a Swine Model of Metabolic Syndrome	The high-cholesterol diet, experimental group were supplemented with daily oral resveratrol (HCRV; 100 mg/kg/day, *n* = 6) for 11 weeks.	In a swine model with metabolic syndrome and chronic myocardial ischemia, we demonstrate that resveratrol supplementation significantly alters the levels of several key proteins involved in metabolism, cell death, and structural remodeling and decreases myocardial fibrosis.	[90]

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
