# Peer review of "The Effects of Resveratrol in the Treatment of Metabolic Syndrome"

_ijms, 2019, doi:10.3390/ijms20030535_

Reviewer 1 Report

The authors reviewed effects of resveratrol in treatment of metabolic syndrome and related diseases. This provides useful information to the researchers and readers.

Major points

1. Page 7 about Liver: How does resveratrol affect to hepatic stellate cells or hepatic fibrosis? The authors should mention this point.

2. Page 8 about Adipose tissue: How does resveratrol affects to immunity, such as M1 or M2 macrophage? The authors should mention this point

Minor points

1. Page 1, line 30: not “trihydroxystilbenen” but “trihydroxystilbene”.

2. Page 4, line 131: How is “CR”? Please full-spell.

Author Response

Major points

1.     Page 7 about Liver: How does resveratrol affect to hepatic stellate cells or hepatic fibrosis? The authors should mention this point.

Thanks for your point. Please see line 227-231. We have added relevant discussion as below.

In rats, N'-Nitrosodimethylamine-induced liver fibrosis was alleviated by resveratrol. Resveratrol was able to revert the liver damages by increasing histological integrity, repressing oxidative damage and inhibiting liver stellate cells activation by down regulating α-Smooth muscle actin expression[63].

2.     Page 8 about Adipose tissue: How does resveratrol affects to immunity, such as M1 or M2 macrophage? The authors should mention this point

Thanks for your point. Please see line 291-302. We have added relevant discussion as below.

Obesity results in chronic inflammation in adipose tissue, and that cells of the innate immune system, particularly macrophages, are crucially involved in adipose inflammation [90]. Macrophages are subdivided into M1 and M2 in general. M1 macrophages are characterized by the production of pro-inflammatory cytokines, while M2 is anti-inflammatory macrophages that involved in the resolution of the inflammatory process and tissue repairing. The previous study has shown that resveratrol exerts anti-inflammatory effects by suppressing M1 microglia activation and promote microglia polarization toward M2 phenotype via PGC-1α [91]. While the role of resveratrol for macrophage polarization seems controversial for adipose tissue. Resveratrol has been shown to attenuate hypoxia-induced macrophage migration to visceral white adipose tissue [92]. However, resveratrol supplementation fails to improve obesity-induced macrophage infiltration and switching [93]. Thus, the effect of resveratrol on macrophage polarization in adipose tissue may be mechanism-dependent.

Minor points

1.     Page 1, line 30: not “trihydroxystilbenen” but “trihydroxystilbene”.

Thanks. Corrected, Line 30

2.     Page 4, line 131: How is “CR”? Please full-spell.

Thanks. Corrected as calorie restriction, Page 4, Line 102

Li-Tung Huang, MD

Department of Pediatrics, Kaohsiung Chang Gung Memorial Hospital, and Chang Gung University College of Medicine, Kaohsiung 833, Taiwan

Reviewer 2 Report

This is a review on the effects of resveratrol in the treatment of metabolic syndrome and its related diseases. The authors first provided sufficient introduction on the natural occurrence, biosynthesis and beneficial activities of resveratrol, then specifically focused on the beneficial activities and underlying mechanisms of resveratrol against metabolic syndrome.

Concerns and suggestions:

1.       Species names should be in italic font, such as Veratrum grandiflorum, Polygonum cuspidatum and Saccharomyces cerevisiae.

2.       Line 57, in vivo should be in vivo.

3.       Line 61, “through” should be “though”?

4.       Sections 2 and 3 should be combined with introduction. Some contents were repeatedly presented, such as the various beneficial activities of resveratrol (line 51-54; line 85-90; line 194-196). Such redundancy should be avoided.

5.       Section 5 seems to be unnecessary considering that similar mechanisms of action were presented in section 6 when different studies were summarized. For instance, the SIRT1 target was mentioned in the liver studies in 6.2.2, AMPK pathway was mentioned as the target in studies focused on kidney.

Author Response

1.      Species names should be in italic font, such as Veratrum grandiflorumPolygonum cuspidatum and Saccharomyces cerevisiae.

Thanks. Corrected, Line 37, 38, 42, 45

2.      Line 57, in vivo should be in vivo.

Thanks. we have merged section 2 and 3 into one section.

3.      Line 61, “through” should be “though”?

Thanks. we have merged section 2 and 3 into one section.

4.       Sections 2 and 3 should be combined with introduction. Some contents were repeatedly presented, such as the various beneficial activities of resveratrol (line 51-54; line 85-90; line 194-196). Such redundancy should be avoided.

Thanks for your point. We shortened the introduction. Also, we have merged section 2 and 3 into one section and avoid redundancy.

5.       Section 5 seems to be unnecessary considering that similar mechanisms of action were presented in section 6 when different studies were summarized. For instance, the SIRT1 target was mentioned in the liver studies in 6.2.2, AMPK pathway was mentioned as the target in studies focused on kidney.

Thanks for your point. We condense our section 5. We prefer to preserve section 5 to have a better framework and to correspond with Fig 2.

Li-Tung Huang, MD

Department of Pediatrics, Kaohsiung Chang Gung Memorial Hospital, and Chang Gung University College of Medicine, Kaohsiung 833, Taiwan

Round  2

Reviewer 1 Report

The authors have responded to the previous concerns.

Line297: Please add "in conditions of neuroinflammatory injury." 

Reviewer 2 Report

The authors have properly addressd the previous concerns in the revised manuscript. I recommend this manuscript to be published in IJMS.